
# Matrix Depot: an extensible test matrix collection for Julia

Weijian Zhang and Nicholas J. Higham

School of Mathematics, University of Manchester, Manchester, UK

## ABSTRACT

Matrix Depot is a Julia software package that provides easy access to a large and diverse collection of test matrices. Its novelty is threefold. First, it is extensible by the user, and so can be adapted to include the user's own test problems. In doing so, it facilitates experimentation and makes it easier to carry out reproducible research. Second, it amalgamates in a single framework two different types of existing matrix collections, comprising parametrized test matrices (including Hansen's set of regularization test problems and Higham's Test Matrix Toolbox) and real-life sparse matrix data (giving access to the University of Florida sparse matrix collection). Third, it fully exploits the Julia language. It uses multiple dispatch to help provide a simple interface and, in particular, to allow matrices to be generated in any of the numeric data types supported by the language.

## INTRODUCTION

In 1969, Gregory and Karney published a book of test matrices (*Gregory & Karney, 1969*). They stated that "In order to test the accuracy of computer programs for solving numerical problems, one needs numerical examples with known solutions. The aim of this monograph is to provide the reader with suitable examples for testing algorithms for finding the inverses, eigenvalues, and eigenvectors of matrix." At that time it was common for journal papers to be devoted to introducing and analyzing a particular test matrix or class of matrices, examples being the papers of *Clement (1959)* (in the first issue of SIAM Review), *Pei (1962)* (occupying just a quarter of a page), and *Gear (1969)*.

Today, test matrices remain of great interest, but not for the same reasons as fifty years ago. Testing accuracy using problems with known solutions is less common because a reference solution correct to machine precision can usually be computed at higher precision without difficulty. The main uses of test matrices nowadays are for exploring the behavior of mathematical quantities (such as eigenvalue bounds) and for measuring the performance of one or more algorithms with respect to accuracy, stability, convergence rate, speed, or robustness.

Various collections of matrices have been made available in software. As well as giving easy access to matrices these collections have the advantage of facilitating reproducibility of experiments (*Donoho & Stodden, 2015*), whether by the same researcher months later or by different researchers.

Corresponding author
Nicholas J. Higham,
nick.higham@manchester.ac.uk

An early collection of parametrizable matrices was given by *Higham (1991)* and made available in MATLAB form. The collection was later extended and distributed as a MATLAB toolbox (*Higham, 1995*). Many of the matrices in the toolbox were subsequently incorporated into the MATLAB `gallery` function. Marques, Vömel, Demmel, and Parlett (*Marques et al., 2008*) present test matrices for tridiagonal eigenvalue problems (already recognized as important by Gregory and Karney, who devoted the last chapter of their book to such matrices). The Harwell–Boeing collection of sparse matrices (*Duff, Grimes & Lewis, 1989*) has been widely used, and is incorporated in the University of Florida Sparse Matrix Collection[1] (*Davis & Hu, 2011*), which contains over 2700 matrices from practical applications, including standard and generalized eigenvalue problems from *Bai et al. (1997)*. Among other MATLAB toolboxes we mention the CONTEST toolbox (*Taylor & Higham, 2009*), which produces adjacency matrices describing random networks, and the NLEVP collection of nonlinear eigenvalue problems (*Betcke et al., 2013*).

The purpose of this work is to provide a test matrix collection for Julia (*Bezanson et al., 2014*; *Bezanson et al., 2012*), a new dynamic programming language for technical computing. The collection, called Matrix Depot, exploits Julia's multiple dispatch features to enable all matrices to be accessed by one simple interface. Moreover, Matrix Depot is extensible. Users can add matrices from the University of Florida Sparse Matrix Collection and Matrix Market; they can code new matrix generators and incorporate them into Matrix Depot; and they can define new groups of matrices that give easy access to subsets of matrices. The parametrized matrices can be generated in any appropriate numeric data type, such as

- floating-point types `Float16` (half precision: 16 bits), `Float32` (single precision: 32 bits), and `Float64` (double precision: 64 bits);
- integer types `Int32` (signed 32-bit integers), `UInt32` (unsigned 32-bit integers), `Int64` (signed 64-bit integers), and `UInt64` (unsigned 64-bit integers);
- `Complex`, where the real and imaginary parts are of any `Real` type (the same for both);
- `Rational` (ratio of integers); and
- arbitrary precision type `BigFloat` (with default precision 256 bits), which uses the GNU MPFR Library (*Fousse et al., 2007*).

This paper is organized as follows. We start by giving a brief demonstration of Matrix Depot in 'A Taste of Matrix Depot.' Then we explain the design and implementation of Matrix Depot in 'Package Design and Implementation,' giving details on how multiple dispatch is exploited; how the collection is stored, accessed, and documented; and how it can be extended. In 'The Matrices' we describe the two classes of matrices in Matrix Depot: parametrized test matrices and real-life sparse matrix data. Concluding remarks are given in the final section.

[1] The University of Florida Sparse Matrix Collection is is to be renamed as The SuiteSparse Matrix Collection.

# A TASTE OF MATRIX DEPOT

To download Matrix Depot, in a Julia REPL (read-eval-print loop) run the command

```
> Pkg.add("MatrixDepot")
```

Then import Matrix Depot into the local scope.

```
> using MatrixDepot
```

Now the package is ready to be used. First, we find out what matrices are in Matrix Depot.

```
> matrixdepot()

Matrices:
    1) baart         2) binomial     3) blur         4) cauchy
    5) chebspec      6) chow         7) circul       8) clement
    9) companion    10) deriv2      11) dingdong    12) fiedler
   13) forsythe     14) foxgood     15) frank       16) golub
   17) gravity      18) grcar       19) hadamard    20) hankel
   21) heat         22) hilb        23) invhilb     24) invol
   25) kahan        26) kms         27) lehmer      28) lotkin
   29) magic        30) minij       31) moler       32) neumann
   33) oscillate    34) parallax    35) parter      36) pascal
   37) pei          38) phillips    39) poisson     40) prolate
   41) randcorr     42) rando       43) randsvd     44) rohess
   45) rosser       46) sampling    47) shaw        48) spikes
   49) toeplitz     50) tridiag     51) triw        52) ursell
   53) vand         54) wathen      55) wilkinson   56) wing

Groups:
  all           data          eigen         ill-cond
  inverse       pos-def       random        regprob
  sparse        symmetric
```

All the matrices and groups in the collection are shown. It is also possible to obtain just the list of matrix names.

```
> matrixdepot("all")
56-element Array{ASCIIString,1}:
 "baart"
 "binomial"
 "blur"
 "cauchy"
 "chebspec"
 "chow"
 "circul"
 "clement"
 "companion"
 "deriv2"
  ...
 "spikes"
 "toeplitz"
 "tridiag"
 "triw"
 "ursell"
 "vand"
 "wathen"
```

```
    "wilkinson"
    "wing"
```

Here, "..." denotes that we have omitted some of the output in order to save space. Next, we check the input options of the Hilbert matrix `hilb`.

```
> matrixdepot("hilb")
     Hilbert matrix
     ================

  The Hilbert matrix has (i,j) element 1/(i+j-1). It is notorious
  for being ill conditioned. It is symmetric positive definite
  and totally positive.

  Input options:

    *  [type,] dim: the dimension of the matrix.

    *  [type,] row_dim, col_dim: the row and column dimensions.

  Groups: ["inverse", "ill-cond", "symmetric", "pos-def"]

  References:

  M. D. Choi, Tricks or treats with the Hilbert matrix, Amer. Math.
  Monthly, 90 (1983), pp. 301-312.

  N. J. Higham, Accuracy and Stability of Numerical Algorithms,
  second  edition, Society for Industrial and Applied Mathematics,
  Philadelphia, PA,  USA, 2002; sec. 28.1.
```

Note that an optional first argument `type` can be given; it defaults to `Float64`. The string of equals signs on the third line in the output above is Markdown notation for a header. Julia interprets Markdown within documentation, though as we are using typewriter font for code examples here, we display the uninterpreted source. We generate a $4 \times 6$ Hilbert matrix with elements in the default double precision type and then in `Rational` type.

```
> matrixdepot("hilb", 4, 6)
4x6 Array{Float64,2}:
 1.0       0.5       0.333333  0.25      0.2       0.166667
 0.5       0.333333  0.25      0.2       0.166667  0.142857
 0.333333  0.25      0.2       0.166667  0.142857  0.125
 0.25      0.2       0.166667  0.142857  0.125     0.111111

> matrixdepot("hilb", Rational, 4, 6)
4x6 Array{Rational{T<:Integer},2}:
 1//1  1//2  1//3  1//4  1//5  1//6
 1//2  1//3  1//4  1//5  1//6  1//7
 1//3  1//4  1//5  1//6  1//7  1//8
 1//4  1//5  1//6  1//7  1//8  1//9
```

A list of all the symmetric matrices in the collection is readily obtained.

```
> matrixdepot("symmetric")
21-element Array{ASCIIString,1}:
 "cauchy"
```

```
"circul"
"clement"
"dingdong"
"fiedler"
"hankel"
"hilb"
"invhilb"
"kms"
"lehmer"
"minij"
"moler"
"oscillate"
"pascal"
"pei"
"poisson"
"prolate"
"randcorr"
"tridiag"
"wathen"
"wilkinson"
```

Here, `symmetric` is one of several predefined groups, and multiple groups can be intersected. For example, the `for` loop below prints the smallest and largest eigenvalues of all the $4 \times 4$ matrices in Matrix Depot that are symmetric positive definite and (potentially) ill conditioned.

```
> for name in matrixdepot("symmetric", "pos-def", "ill-cond")
      A = full(matrixdepot(name, 4))
      @printf "
      name eigmin(A) eigmax(A)
  end

   cauchy: smallest eigval = 2.131e-05, largest eigval = 9.776e-01
     hilb: smallest eigval = 9.670e-05, largest eigval = 1.500e+00
  invhilb: smallest eigval = 6.666e-01, largest eigval = 1.034e+04
      kms: smallest eigval = 3.750e-01, largest eigval = 2.086e+00
    moler: smallest eigval = 3.336e-02, largest eigval = 5.122e+00
oscillate: smallest eigval = 1.490e-08, largest eigval = 1.000e+00
   pascal: smallest eigval = 3.802e-02, largest eigval = 2.630e+01
      pei: smallest eigval = 1.000e+00, largest eigval = 5.000e+00
  tridiag: smallest eigval = 3.820e-01, largest eigval = 3.618e+00
```

Matrices can also be accessed by number within the alphabetical list of matrix names.

```
> matrixdepot(2)
"binomial"

> matrixdepot(2:5)
4-element Array{AbstractString,1}:
 "binomial"
 "blur"
 "cauchy"
 "chebspec"

> matrixdepot(15:20, 5, 6, 1:3)
11-element Array{AbstractString,1}:
 "frank"
 "golub"
 "gravity"
 "grcar"
```

```
"hadamard"
"hankel"
"chebspec"
"chow"
"baart"
"binomial"
"blur"
```

Access by number provides a convenient way to run a test on subsets of matrices in the collection. However, the number assigned to a matrix may change if we include new matrices in the collection. In order to run tests in a way that is repeatable in the future it is best to group matrices into subsets using the macro `@addgroup`, which stores them by name. For example, the following command will group test matrices `frank`, `golub`, `gravity`, `grcar`, `hadamard`, `hankel`, `chebspec`, `chow`, `baart`, `binomial`, and `blur` into `test1`.

```
> @addgroup test1 = matrixdepot(15:20, 5, 6, 1:3)
```

After reloading the package, we can run tests on these matrices using group `test1`. Here we compute the 2-norms. Since `blur` (an image deblurring test problem) generates a sparse matrix and the matrix 2-norm is currently not implemented for sparse matrices in Julia, we use `full` to convert the matrix to dense format.

```
> for name in matrixdepot("test1")
      A = full(matrixdepot(name , 4))
      @printf "\%9s has 2-norm \%0.3e \n" name norm(A)
  end

    baart has 2-norm 3.192e+00
 binomial has 2-norm 4.576e+00
     blur has 2-norm 8.298e-01
 chebspec has 2-norm 6.474e+00
      chow has 2-norm 3.414e+00
     frank has 2-norm 7.624e+00
     golub has 2-norm 2.050e+02
   gravity has 2-norm 6.656e+00
     grcar has 2-norm 2.562e+00
  hadamard has 2-norm 2.000e+00
    hankel has 2-norm 1.160e+01
```

To download the test matrix `SNAP/web-Google` from the University of Florida Sparse Matrix Collection (see 'Matrix Data from External Sources' for more details), we first download the data with

```
> matrixdepot("SNAP/web-Google", :get)
```

and then generate the matrix with

```
> matrixdepot("SNAP/web-Google", :r)
916428x916428 sparse matrix with 5105039 Float64 entries:
 [11343 ,       1]  =  1.0
 [11928 ,       1]  =  1.0
 [15902 ,       1]  =  1.0
 [29547 ,       1]  =  1.0
```

```
[30282 ,      1]  =  1.0
[31301 ,      1]  =  1.0
[38717 ,      1]  =  1.0
...
[720325, 916427]  =  1.0
[772226, 916427]  =  1.0
[785097, 916427]  =  1.0
[788476, 916427]  =  1.0
[822938, 916427]  =  1.0
[833616, 916427]  =  1.0
[417498, 916428]  =  1.0
[843845, 916428]  =  1.0
```

Note that the omission marked " ... " was in this case automatically done by Julia based on the height of the terminal window. Matrices loaded in this way are inserted into the list of available matrices, and assigned a number. After downloading further matrices HB/1138_bus, HB/494_bus, and Bova/rma10 the list of matrices is as follows.

```
julia> matrixdepot()

Matrices:
  1) baart      2) binomial    3) blur       4) cauchy
  5) chebspec   6) chow        7) circul     8) clement
  9) companion 10) deriv2     11) dingdong  12) fiedler
 13) forsythe  14) foxgood    15) frank     16) golub
 17) gravity   18) grcar      19) hadamard  20) hankel
 21) heat      22) hilb       23) invhilb   24) invol
 25) kahan     26) kms        27) lehmer    28) lotkin
 29) magic     30) minij      31) moler     32) neumann
 33) oscillate 34) parallax   35) parter    36) pascal
 37) pei       38) phillips   39) poisson   40) prolate
 41) randcorr  42) rando      43) randsvd   44) rohess
 45) rosser    46) sampling   47) shaw      48) spikes
 49) toeplitz  50) tridiag    51) triw      52) ursell
 53) vand      54) wathen     55) wilkinson 56) wing
 57) Bova/rma10 58) HB/1138_bus 59) HB/494_bus  60) SNAP/web-Google

Groups:
  all          data         eigen        ill-cond
  inverse      pos-def      random       regprob
  sparse       symmetric    test1
```

## PACKAGE DESIGN AND IMPLEMENTATION

In this section we describe the design and implementation of Matrix Depot, focusing particularly on the novel aspects of exploitation of multiple dispatch, extensibility of the collection, and user-definable grouping of matrices.

### Exploiting multiple dispatch

Matrix Depot makes use of multiple dispatch in Julia, an object-oriented paradigm in which the selection of a function implementation is based on the types of each argument of the function. The generic function matrixdepot has eight different methods, where each method itself is a function that handles a specific case. This is neater and more convenient than writing eight "case" statements, as is necessary in many other languages.

```
> methods(matrixdepot)
# 8 methods for generic function "matrixdepot":
matrixdepot() ...
matrixdepot(name::AbstractString) ...
matrixdepot(name::AbstractString, method::Symbol) ...
matrixdepot(props::AbstractString...) ...
matrixdepot(name::AbstractString, args...) ...
matrixdepot(num::Integer) ...
matrixdepot(ur::UnitRange{T<:Real}) ...
matrixdepot(vs::Union{Integer,UnitRange{T<:Real}}...) ...
```

For example, the following two functions are used for accessing matrices by number and range respectively, where `matrix_name_list()` returns a list of matrix names. The second function calls the first function in the inner loop.

```
function matrixdepot(num::Integer)
    matrixstrings = matrix_name_list()
    n = length(matrixstrings)
    if num > n
        error("There are $(n) parameterized matrices,
                but you asked for the $(num)-th.")
    end
    return matrixstrings[num]
end

function matrixdepot(ur::UnitRange)
    matrixnamelist = AbstractString[]
    for i in ur
        push!(matrixnamelist, matrixdepot(i))
    end
    return matrixnamelist
end
```

As a result, `matrixdepot` is a versatile function that can be used for a variety of purposes, including returning matrix information and generating matrices from various input parameters.

In the following example we see how multiple dispatch handles different numbers and types of arguments for the Cauchy matrix.

```
> matrixdepot("cauchy")
    Cauchy matrix
    =============

  Given two vectors x and y, the (i,j) entry of the Cauchy matrix
  is  1/(x[i]+y[j]).

  Input options:

    *  [type,] x, y: two vectors.

    *  [type,] x: a vector. y defaults to x.

    *  [type,] dim: the dimension of the matrix. x and y default to
    [1:dim;].

  Groups: ["inverse", "ill-cond", "symmetric", "pos-def"]
```

```
    References:

    N. J. Higham, Accuracy and Stability of Numerical Algorithms,
    second edition, Society for Industrial and Applied Mathematics,
    Philadelphia, PA, USA, 2002; sec. 28.1

> matrixdepot("cauchy", [1, 2, 3], [4, 5, 6])
3x3 Array{Float64,2}:
 0.2       0.166667  0.142857
 0.166667  0.142857  0.125
 0.142857  0.125     0.111111

> matrixdepot("cauchy", [0.2, 0.3, 0.4])
3x3 Array{Float64,2}:
 2.5       2.0       1.66667
 2.0       1.66667   1.42857
 1.66667   1.42857   1.25

> matrixdepot("cauchy", 3)
3x3 Array{Float64,2}:
 0.5       0.333333  0.25
 0.333333  0.25      0.2
 0.25      0.2       0.166667

> matrixdepot("cauchy", Float32, 3)
3x3 Array{Float32,2}:
 0.5       0.333333  0.25
 0.333333  0.25      0.2
 0.25      0.2       0.166667
```

Multiple dispatch is also exploited in programming the matrices. For example, the Hilbert matrix is implemented as

```
function hilb{T}(::Type{T}, m::Integer, n::Integer)
    H = zeros(T, m, n)
    for j = 1:n, i = 1:m
        @inbounds H[i,j] = one(T)/ (i + j - one(T))
    end
    return H
end
hilb{T}(::Type{T}, n::Integer) = hilb(T, n, n)
hilb(args...) = hilb(Float64, args...)
```

The function `hilb` has three methods, which enable one to request, for example, `hilb(4,2)` for a $4 \times 2$ Hilbert matrix of type `Float64`, or simply (thanks to the final two lines) `hilb(4)` for a $4 \times 4$ Hilbert matrix of type `Float64`. The keyword `@inbounds` tells Julia to turn off bounds checking in the following expression, in order to speed up execution. Note that in Julia it is not necessary to vectorize code to achieve good performance (*Bezanson et al., 2014*).

All the matrices in Matrix Depot can be generated using the function call

```
matrixdepot("matrix_name", p1, p2, ...),
```

where `matrix_name` is the name of the test matrix, and `p1, p2, ...,` are input arguments depending on `matrix_name`. The help comments for each matrix can be viewed by

calling function `matrixdepot("matrix_name")`. We can access the list of matrix names by number, range, or a mixture of numbers and range.

1. `matrixdepot(i)` returns the name of the $i$th matrix;
2. `matrixdepot(i:j)` returns the names of the $i$th to $j$th matrices, where $i < j$;
3. `matrixdepot(i:j, k, m)` returns the names of the $i$th, $(i+1)$st, ..., $j$th, $k$th, and $m$th matrices.

## Matrix representation

Matrix names in Matrix Depot are represented by Julia strings. For example, the Cauchy matrix is represented by `"cauchy"`. Matrix names and matrix groups are stored as hash tables (`Dict`). In particular, there is a hash table `matrixdict` that maps each matrix name to its underlying function and a hash table `matrixclass` that maps each group to its members.

The majority of parametrized matrices are dense matrices of type `Array{T,2}`, where `T` is the element type of the matrix. Variables of the `Array` type are stored in column-major order. A few matrices are stored as sparse matrices (see also `matrixdepot("sparse")`), in the Compressed Sparse Column (CSC) format; these include `neumann` (a singular matrix from the discrete Neumann problem) and `poisson` (a block tridiagonal matrix from Poisson's equation). Tridiagonal matrices are stored in the built-in Julia type `Tridiagonal`, which is defined as follows.

```
immutable Tridiagonal{T} <: AbstractMatrix{T}
    dl::Vector{T}    # sub-diagonal
    d::Vector{T}     # diagonal
    du::Vector{T}    # sup-diagonal
    du2::Vector{T}   # supsup-diagonal for pivoting
end
```

## Matrix groups

A group is a subset of matrices in Matrix Depot. There are ten predefined groups, described in Table 1, most of which identify matrices with particular properties. Each group is represented by a string. For example, the group of random matrices is represented by `"random"`. Matrices can be accessed by group names, as was illustrated in 'A Taste of Matrix Depot.'

The macro `@addgroup` is used to add a new group of matrices to Matrix Depot and the macro `@rmgroup` removes an added group. All the predefined matrix groups are stored in the hash table `matrixclass`. The macro `addgroup` essentially adds a new key-value combination to the hash table `usermatrixclass`. Using a separate hash table prevents the user from contaminating the predefined matrix groups.

Being able to create groups is a useful feature for reproducible research (*Donoho & Stodden, 2015*). For example, if we have implemented algorithm `alg01` and we used `circul`, `minij`, and `grcar` as test matrices for `alg01`, we could type

```
> @addgroup alg01_group = ["circul", "minij", "grcar"]
```

**Table 1  Predefined groups.**

| Group | Description |
| --- | --- |
| all | All the matrices in the collection. |
| data | The matrix has been downloaded from the University of Florida Sparse Collection or the Matrix Market Collection. |
| eigen | Part of the eigensystem of the matrix is explicitly known. |
| ill-cond | The matrix is ill-conditioned for some parameter values. |
| inverse | The inverse of the matrix is known explicitly. |
| pos-def | The matrix is positive definite for some parameter values. |
| random | The matrix has random entries. |
| regprob | The output is a test problem for regularization methods. |
| sparse | The matrix is sparse. |
| symmetric | The matrix is symmetric for some parameter values. |

This adds a new group to Matrix Depot (we need to reload the package to see the changes).

```
julia> matrixdepot()

Matrices:
   1) baart        2) binomial     3) blur         4) cauchy
   5) chebspec     6) chow         7) circul       8) clement
   9) companion   10) deriv2      11) dingdong    12) fiedler
  13) forsythe    14) foxgood     15) frank       16) golub
  17) gravity     18) grcar       19) hadamard    20) hankel
  21) heat        22) hilb        23) invhilb     24) invol
  25) kahan       26) kms         27) lehmer      28) lotkin
  29) magic       30) minij       31) moler       32) neumann
  33) oscillate   34) parallax    35) parter      36) pascal
  37) pei         38) phillips    39) poisson     40) prolate
  41) randcorr    42) rando       43) randsvd     44) rohess
  45) rosser      46) sampling    47) shaw        48) spikes
  49) toeplitz    50) tridiag     51) triw        52) ursell
  53) vand        54) wathen      55) wilkinson   56) wing

Groups:
  all           data           eigen          ill-cond
  inverse       pos-def        random         regprob
  sparse        symmetric      alg01_group
```

We can then run `alg01` on the test matrices by

```
> for name in matrixdepot(alg01_group)
     A = matrixdepot(name, n)  # n is the dimension of the matrix.
     @printf "Test result for
  end
```

## Adding new matrix generators

Generators are Julia functions that generate test matrices. When Matrix Depot is first loaded, a directory `myMatrixDepot` is created. It contains two files, `group.jl` and `generator.jl`, where `group.jl` is used for storing all the user-defined groups (see 'Matrix Group') and `generator.jl` is used for storing generator declarations.

[2]Git is a free and open source distributed
version control system.

Julia packages are simply Git repositories.[2] The directory `myMatrixDepot` is untracked by Git, so any local changes to files in `myMatrixDepot` do not make the `MatrixDepot` package "dirty." In particular, all the newly defined groups or matrix generators will not be affected when we upgrade to a new version of Matrix Depot. Matrix Depot automatically loads all Julia files in `myMatrixDepot`. This feature allows a user to simply drop generator files into `myMatrixDepot` without worrying about how to link them to Matrix Depot.

A new generator is declared using the syntax `include_generator(FunctionName,` `"fname", f)`. This adds the new mapping `"fname"` → `f` to the hash table `matrixdict`, which we recall maps each matrix name to its underlying function. Matrix Depot will refer to function `f` using string `"fname"` so that we can call function `f` by `matrixdepot("fname"...)`. The user is free to define new data types and return values of those types. Moreover, as with any Julia function, multiple values can be returned by listing them after the `return` statement.

For example, suppose we have the following Julia file `rand.jl`, which contains two generators `randsym` and `randorth` and we want to use them from Matrix Depot. The triple quotes in the file delimit the documentation for the functions.

```
"""
random symmetric matrix
=======================

Input options:

* n: the dimension of the matrix
"""
function randsym(n)
  A = zeros(n, n)
  for j = 1:n
      for i = 1:j
          A[i,j] = randn()
          if i != j; A[j,i] = A[i,j] end
      end
  end
  return A
end

"""
random orthogonal matrix
========================

Input options:

* n: the dimension of the matrix
"""
randorth(n) = qr(randn(n,n))[1]
```

We can copy the file `rand.jl` to the directory `myMatrixDepot` and add the following two lines to `generator.jl`.

```
include_generator(FunctionName, "randsym", randsym)
include_generator(FunctionName, "randorth", randorth)
```

This includes the functions `randsym` and `randorth` in Matrix Depot, as we can see by looking at the matrix list (the new entries are numbered 43 and 45).

```
julia> matrixdepot()

Matrices:
   1) baart          2) binomial       3) blur           4) cauchy
   5) chebspec       6) chow           7) circul         8) clement
   9) companion     10) deriv2        11) dingdong      12) fiedler
  13) forsythe      14) foxgood       15) frank         16) golub
  17) gravity       18) grcar         19) hadamard      20) hankel
  21) heat          22) hilb          23) invhilb       24) invol
  25) kahan         26) kms           27) lehmer        28) lotkin
  29) magic         30) minij         31) moler         32) neumann
  33) oscillate     34) parallax      35) parter        36) pascal
  37) pei           38) phillips      39) poisson       40) prolate
  41) randcorr      42) rando         43) randorth      44) randsvd
  45) randsym       46) rohess        47) rosser        48) sampling
  49) shaw          50) spikes        51) toeplitz      52) tridiag
  53) triw          54) ursell        55) vand          56) wathen
  57) wilkinson     58) wing
Groups:
  all            data           eigen          ill-cond
  inverse        pos-def        random         regprob
  sparse         symmetric
```

The new generators can be used just like the built-in ones.

```
> matrixdepot("randsym")
     random symmetric matrix
     =======================

    Input options:

      *  n: the dimension of the matrix

> matrixdepot("randsym", 4)
4x4 Array{Float64,2}:
 -0.00992523  0.174531   -1.73322    -0.765096
  0.174531    1.69308     0.269062    0.594058
 -1.73322     0.269062   -0.824277   -0.541458
 -0.765096    0.594058   -0.541458   -0.480428

> matrixdepot("randorth")
     random orthogonal matrix
     ========================

    Input options:

      *  n: the dimension of the matrix

> A = matrixdepot("randorth", 4)
4x4 Array{Float64,2}:
 -0.233943   0.179893    0.563926   -0.771295
 -0.769649  -0.141938   -0.5807     -0.224235
  0.247165   0.832118   -0.449941   -0.20986
 -0.540204   0.505046    0.377263    0.557477

> A'*A - eye(4,4)
4x4 Array{Float64,2}:
 -2.22045e-16   1.66533e-16  -2.77556e-17  -1.66533e-16
```

```
  1.66533e-16  -1.11022e-16  -3.05311e-16   1.66533e-16
 -2.77556e-17  -3.05311e-16  -1.11022e-16   1.94289e-16
 -1.66533e-16   1.66533e-16   1.94289e-16   0.0
```

We can also add group information with the function `include_generator`. The following lines are put in `generator.jl`.

```
include_generator(Group, "random", randsym)
include_generator(Group, "random", randorth)
```

This adds the functions `randsym` and `randorth` to the group `random`, as we can see with the following query (after reloading the package).

```
> matrixdepot("random")
10-element Array{ASCIIString,1}:
 "golub"
 "oscillate"
 "randcorr"
 "rando"
 "randorth"
 "randsvd"
 "randsym"
 "rohess"
 "rosser"
 "wathen"
```

### Documentation

The Matrix Depot documentation is created using the documentation generator Sphinx (http://sphinx-doc.org/) and is hosted at Read the Docs (http://matrixdepotjl.readthedocs.org). Its primary goals are to provide examples of usage of Matrix Depot and to give a brief summary of each matrix in the collection. Matrices are listed alphabetically with hyperlinks to the documentation for each matrix. Most parametrized matrices are presented with heat map plots, which are produced using the Winston package (https://github.com/nolta/Winston.jl), with the color range determined by the smallest and largest entries of the matrix. For example, Fig. 1 shows how the Wathen matrix is documented in Matrix Depot.

## THE MATRICES

We now describe the matrices that are provided with, or can be downloaded into, Matrix Depot.

### Parametrized matrices

In Matrix Depot v0.5.5, there are 58 parametrized matrices (including the regularization problems described in the next section), most of which originate from the Test Matrix Toolbox (*Higham, 1995*). All these matrices can be generated as `matrixdepot("matrix_name", n)`, where n is the dimension of the matrix.

Many matrices can have more than one input parameter, and multiple dispatch provides a convenient mechanism for taking different actions for different argument types. For

**wathen**

The Wathen matrix is a sparse, symmetric positive, random matrix arising from the finite element method [wath87]. It is the consistent mass matrix for a regular *nx-by-ny* grid of 8-node elements.

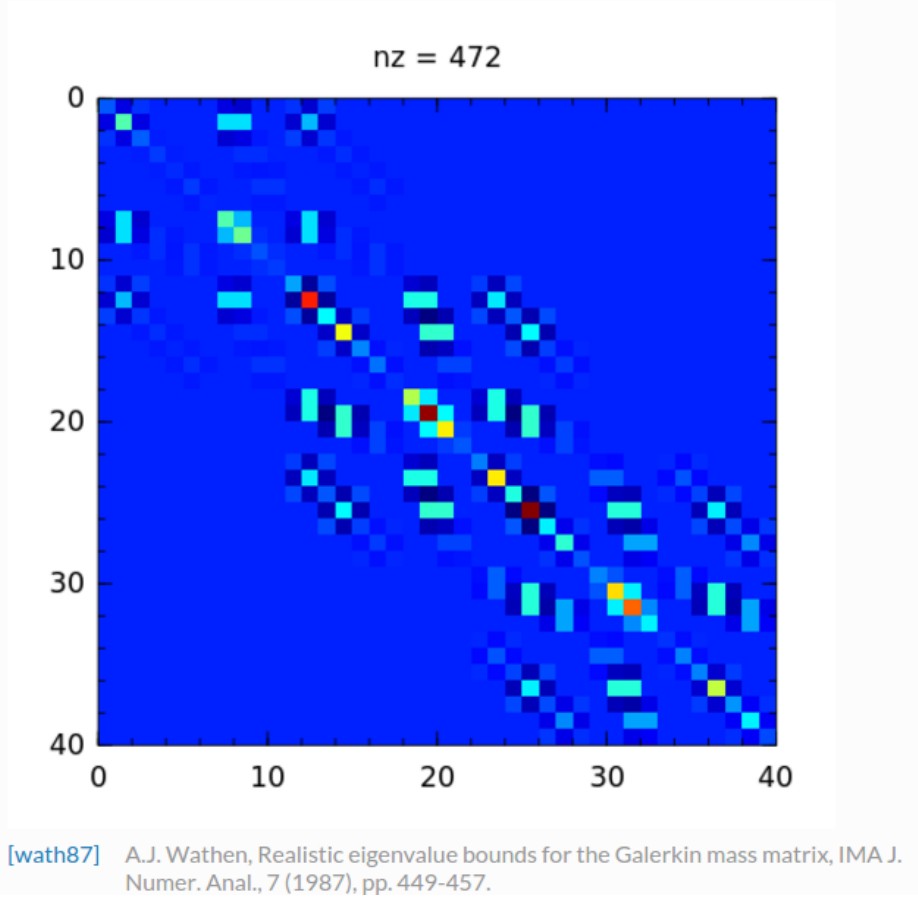

[wath87]   A.J. Wathen, Realistic eigenvalue bounds for the Galerkin mass matrix, IMA J. Numer. Anal., 7 (1987), pp. 449-457.

**Figure 1   Documentation for the Wathen matrix.**

example, the `tridiag` function generates a tridiagonal matrix from vector arguments giving the subdiagonal, diagonal, and superdiagonal vectors, but a tridiagonal Toeplitz matrix can be obtained by supplying scalar arguments that specify the dimension of the matrix, the subdiagonal, the diagonal, and the superdiagonal. If a single, scalar argument n is supplied then an n-by- n tridiagonal Toeplitz matrix with subdiagonal and superdiagonal −1 and diagonal 2 is constructed. This matrix arises in applying central differences to a second derivative operator, and the inverse and the condition number are known explicitly (*Higham, 2002*, sec. 28.5).

Here is an example of the different usages of `tridiag`.

```
> matrixdepot("tridiag")
     Tridiagonal Matrix
     ===================

  Construct a tridiagonal matrix of type Tridiagonal.
```

```
Input options:

   *  [type,] v1, v2, v3: v1 and v3 are vectors of subdiagonal and
      superdiagonal elements, respectively, and v2 is a vector of
      diagonal elements.

   *  [type,] dim, x, y, z: dim is the dimension of the matrix, x,
   y, z are  scalars. x and z are the subdiagonal and
   superdiagonal elements,
      respectively, and y is the diagonal elements.

   *  [type,] dim: x = -1, y = 2, z = -1. This matrix is also
   known as the  second difference matrix.

  Groups: ["inverse", "ill-cond", "pos-def", "eigen"]

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

> matrixdepot("tridiag", [2,5,6;], ones(4), [3,4,1;])
4x4 Tridiagonal{Float64}:
 1.0  3.0  0.0  0.0
 2.0  1.0  4.0  0.0
 0.0  5.0  1.0  1.0
 0.0  0.0  6.0  1.0

> matrixdepot("tridiag", 4, 5, 3, 1)
4x4 Tridiagonal{Float64}:
 3.0  1.0  0.0  0.0
 5.0  3.0  1.0  0.0
 0.0  5.0  3.0  1.0
 0.0  0.0  5.0  3.0

> matrixdepot("tridiag", Int, 4)
4x4 Tridiagonal{Int64}:
  2  -1   0   0
 -1   2  -1   0
  0  -1   2  -1
  0   0  -1   2
```

### Test problems for regularization methods

A mathematical problem is ill-posed if the solution is not unique or if an arbitrarily small perturbation of the data can cause an arbitrarily large change in the solution. Regularization methods are an important class of methods for dealing with such problems (*Hansen, 1998*; *Hansen, 2010*). One means of generating test problems for regularization methods is to discretize a given ill-posed problem.

Matrix Depot contains a group of regularization test problems derived from Hansen's MATLAB Regularization Tools (*Hansen, 1994*; *Hansen, 2007*; *Hansen, 2008*) that are mostly discretizations of Fredholm integral equations of the first kind:

$$\int_0^1 K(s,t)f(t)\,dt = g(s), \quad 0 \leq s \leq 1.$$

The regularization test problems form the group `regprob`.

```
> matrixdepot("regprob")
12-element Array{ASCIIString,1}:
 "baart"
 "blur"
 "deriv2"
 "foxgood"
 "gravity"
 "heat"
 "parallax"
 "phillips"
 "shaw"
 "spikes"
 "ursell"
 "wing"
```

Each problem is a linear system $Ax = b$ where the matrix $A$ and vectors $x$ and $b$ are obtained by discretization (using quadrature or the Galerkin method) of $K$, $f$, and $g$. By default, we generate only $A$, which is an ill-conditioned matrix. The whole test problem will be generated if the parameter `matrixonly` is set to `false`, and in this case the output has type `RegProb`, which is defined as

```
immutable RegProb{T}
 A::AbstractMatrix{T} # matrix of interest
 b::AbstractVector{T} # right-hand side
 x::AbstractVector{T} # the solution to Ax = b
end
```

If `r` is a generated test problem, then `r.A`, `r.b`, and `r.x` are the matrix $A$ and vectors $x$ and $b$ respectively. If the solution is not provided by the problem, the output is stored as type `RegProbNoSolution`, which is defined as

```
immutable RegProbNoSolution{T}
 A::AbstractMatrix{T} # matrix of interest
 b::AbstractVector{T} # right-hand side
end
```

For example, the test problem `wing` can be generated as follows.

```
> matrixdepot("wing")
     A Problem with a Discontinuous Solution
     =======================================

  Input options:

     *  [type,] dim, t1, t2, [matrixonly]: the dimension of matrix
     is dim. t1 and t2 are two real scalars such that 0 < t1 < t2
     < 1. If matrixonly = false, the matrix A and vectors b and x
     in the linear system Ax = b will be generated(matrixonly =
     true by default).

     *  [type,] n, [matrixonly]: t1 = 1/3 and t2 = 2/3.

  Groups: ["regprob"]
```

```
   References:

   G. M. Wing, A Primer on Integral Equations of the First Kind,
   Society for  Industrial and Applied Mathematics, 1991, p. 109.

> A = matrixdepot("wing", 4)
4x4 Array{Float64,2}:
 0.031189    0.0921165   0.148804   0.198786
 0.0310674   0.0889342   0.134959   0.164156
 0.0309463   0.085862    0.122403   0.13556
 0.0308257   0.0828958   0.111014   0.111945

> r = matrixdepot("wing", 4, false)
Test problems for Regularization Methods
A:
4x4 Array{Float64,2}:
 0.031189    0.0921165   0.148804   0.198786
 0.0310674   0.0889342   0.134959   0.164156
 0.0309463   0.085862    0.122403   0.13556
 0.0308257   0.0828958   0.111014   0.111945
b:
4-element Array{Float64,1}:
 0.0804953
 0.0751385
 0.0701787
 0.0655842
x:
4-element Array{Float64,1}:
 0.0
 0.5
 0.5
 0.0

> r.x
4-element Array{Float64,1}:
 0.0
 0.5
 0.5
 0.0
```

## Matrix data from external sources

Matrix Depot provides access to matrices from Matrix Market (*Boisvert et al., 1997*) and the University of Florida Sparse Matrix Collection (*Davis & Hu, 2011*), both of which contain many matrices taken from applications. In particular, these sources contain many large, sparse matrices.

Matrix Market and the University of Florida Sparse Matrix Collection both categorize matrices by application domain and the problem source and both provide matrices in Matrix Market Format (*Boisvert, Pozo & Remington, 1996*). These similarities allow us to design a generic interface for both collections. The symbol `:get` (or `:g`) is used for downloading matrices from both collections and the symbol `:read` (or `:r`) is used for reading in matrices already downloaded. Downloaded matrix data is stored on disk in the Matrix Market format and when read into Julia is stored in the type `SparseMatrixCSC`.

`MatrixDepot.update()` downloads the matrix name data files from the two web servers.

```
> MatrixDepot.update()
 \% Total  \% Received \% Xferd  Average Speed   Time    Time     Time  Current
                                 Dload  Upload   Total   Spent    Left  Speed
100 1887k  0 1887k    0        0  97337      0 --:--:--  0:00:19 --:--:--  472k
 \% Total  \% Received \% Xferd  Average Speed   Time    Time     Time  Current
                                 Dload  Upload   Total   Spent    Left  Speed
100 41552  0 41552    0        0   4421      0 --:--:--  0:00:09 --:--:-- 41018
```

The University of Florida Sparse Matrix Collection is divided into matrix groups and the group of a matrix forms part of the full name of the matrix (*Davis & Hu, 2011*). For example, the full name of the matrix `1138_bus` in the Harwell-Boeing Collection is `HB/1138_bus`.

```
> matrixdepot("HB/1138_bus", :get)
  \% Total     \% Received \% Xferd  Average Speed   Time    Time     Time  Current
                                     Dload  Upload   Total   Spent    Left  Speed
 100 19829   100 19829    0        0   2320      0 0:00:08  0:00:08 --:--:-- 49572

> matrixdepot("HB/1138_bus", :read)
1138x1138 Symmetric{Float64,SparseMatrixCSC{Float64,Int64}}:
 1474.78      0.0       0.0    ...    0.0       0.0         0.0     0.0
    0.0       9.13665   0.0           0.0       0.0         0.0     0.0
    0.0       0.0      69.6147        0.0       0.0         0.0     0.0
    0.0       0.0       0.0           0.0       0.0         0.0     0.0
   -9.01713   0.0       0.0           0.0       0.0         0.0     0.0
    0.0       0.0       0.0    ...    0.0       0.0         0.0     0.0
    0.0       0.0       0.0           0.0       0.0         0.0     0.0
    0.0       0.0       0.0           0.0       0.0         0.0     0.0
    0.0       0.0       0.0           0.0       0.0         0.0     0.0
    0.0      -3.40599   0.0           0.0       0.0         0.0     0.0
    ...                                                             ...
    0.0       0.0       0.0           0.0       0.0         0.0     0.0
    0.0       0.0       0.0    ...    0.0     -24.3902      0.0     0.0
    0.0       0.0       0.0           0.0       0.0         0.0     0.0
    0.0       0.0       0.0           0.0       0.0         0.0     0.0
    0.0       0.0       0.0           0.0       0.0         0.0     0.0
    0.0       0.0       0.0          26.5639    0.0         0.0     0.0
    0.0       0.0       0.0    ...    0.0      46.1767      0.0     0.0
    0.0       0.0       0.0           0.0       0.0     10000.0     0.0
    0.0       0.0       0.0           0.0       0.0         0.0   117.647
```

Matrices from the University of Florida Sparse Matrix Collection are stored in `MatrixDepot/data/uf` and they are stored by group (to avoid duplicate names), i.e., one directory per group. Similarly, matrices from Matrix Market are stored in `MatrixDepot/data/mm`. Both directories are untracked by Git. Many matrices in the University of Florida Sparse Matrix Collection contain problem-specific metadata, all of which is downloaded. The metadata is accessed by setting the keyword argument `meta` to `true`. Then instead of returning the matrix, Matrix Depot will return the metadata (including the matrix) as a dictionary. For example, the IMDB movie database

`Pajek/IMDB` has metadata related to actors and movies. The following command stores all the metadata of `Pajek/IMDB` in a variable `r`, where `r["IMDB"]` is the matrix.

```
> r = matrixdepot("Pajek/IMDB", :r, meta = true)
Dict{AbstractString,Any} with 8 entries:
  "IMDB_colname"   => "'La Tata' Castro, Maria Tereza\n'La Veneno'...
  "IMDB_MovieBacon" => 428440x1 Array{Float64,2}
  "IMDB_code"      => "Drama\nShort\nDocumentary\nComedy\nWestern\nFamily...
  "IMDB_KevinBacon" => 1x1 Array{Float64,2}
  "IMDB_ActorBacon" => 896308x1 Array{Float64,2}
  "IMDB_category"  => 428440x1 Array{Float64,2}
  "IMDB"           => 428440x896308 sparse matrix with 3782463 Float64 entries
  "IMDB_year"      => 428440x1 Array{Float64,2}
```

We can download a whole group of matrices from the University of Florida sparse matrix collection using the command `matrixdepot("group name/*", :get)`. The next example downloads all 67 matrices in the Gset group of matrices from random graphs (contributed by Y. Ye) then displays all the matrices in Matrix Depot, including the newly downloaded matrices.

```
> matrixdepot("Gset/*", :get)

Downloading all matrices in group Gset...
  \% Total    \% Received \% Xferd  Average Speed   Time    Time     Time  Current
                                  Dload  Upload   Total   Spent    Left  Speed
100 48083  100 48083    0     0  95388      0 --:--:-- --:--:-- --:--:-- 96166
download:/home/weijian/.julia/v0.4/MatrixDepot/src/../data/uf/Gset/G1.tar.gz
G1/G1.mtx
  \% Total    \% Received \% Xferd  Average Speed   Time    Time     Time  Current
                                  Dload  Upload   Total   Spent    Left  Speed
100 55180  100 55180    0     0  75318      0 --:--:-- --:--:-- --:--:-- 75692
download:/home/weijian/.julia/v0.4/MatrixDepot/src/../data/uf/Gset/G10.tar.gz
G10/G10.mtx
  \% Total    \% Received \% Xferd  Average Speed   Time    Time     Time  Current
                                  Dload  Upload   Total   Spent    Left  Speed
100  5926  100  5926    0     0  23126      0 --:--:-- --:--:-- --:--:-- 23515
download:/home/weijian/.julia/v0.4/MatrixDepot/src/../data/uf/Gset/G11.tar.gz
G11/G11.mtx
  \% Total    \% Received \% Xferd  Average Speed   Time    Time     Time  Current
                                  Dload  Upload   Total   Spent    Left  Speed
100  6349  100  6349    0     0  24223      0 --:--:-- --:--:-- --:--:-- 24608
...

> matrixdepot()

Matrices:
   1) baart          2) binomial       3) blur          4) cauchy
   5) chebspec       6) chow           7) circul        8) clement
   9) companion     10) deriv2        11) dingdong     12) fiedler
  13) forsythe      14) foxgood       15) frank        16) golub
  17) gravity       18) grcar         19) hadamard     20) hankel
  21) heat          22) hilb          23) invhilb      24) invol
  25) kahan         26) kms           27) lehmer       28) lotkin
  29) magic         30) minij         31) moler        32) neumann
  33) oscillate     34) parallax      35) parter       36) pascal
```

```
 37) pei             38) phillips      39) poisson        40) prolate
 41) randcorr        42) rando         43) randsvd        44) rohess
 45) rosser          46) sampling      47) shaw           48) spikes
 49) toeplitz        50) tridiag       51) triw           52) ursell
 53) vand            54) wathen        55) wilkinson      56) wing
 57) Gset/G1         58) Gset/G10      59) Gset/G11       60) Gset/G12
 61) Gset/G13        62) Gset/G14      63) Gset/G15       64) Gset/G16
 65) Gset/G17        66) Gset/G18      67) Gset/G19       68) Gset/G2
 69) Gset/G20        70) Gset/G21      71) Gset/G22       72) Gset/G23
 73) Gset/G24        74) Gset/G25      75) Gset/G26       76) Gset/G27
 77) Gset/G28        78) Gset/G29      79) Gset/G3        80) Gset/G30
 81) Gset/G31        82) Gset/G32      83) Gset/G33       84) Gset/G34
 85) Gset/G35        86) Gset/G36      87) Gset/G37       88) Gset/G38
 89) Gset/G39        90) Gset/G4       91) Gset/G40       92) Gset/G41
 93) Gset/G42        94) Gset/G43      95) Gset/G44       96) Gset/G45
 97) Gset/G46        98) Gset/G47      99) Gset/G48      100) Gset/G49
101) Gset/G5        102) Gset/G50     103) Gset/G51      104) Gset/G52
105) Gset/G53       106) Gset/G54     107) Gset/G55      108) Gset/G56
109) Gset/G57       110) Gset/G58     111) Gset/G59      112) Gset/G6
113) Gset/G60       114) Gset/G61     115) Gset/G62      116) Gset/G63
117) Gset/G64       118) Gset/G65     119) Gset/G66      120) Gset/G67
121) Gset/G7        122) Gset/G8      123) Gset/G9
Groups:
  all         data         eigen        ill-cond
  inverse     pos-def      random       regprob
  sparse      symmetric
```

The full name of a matrix in Matrix Market comprises three parts: the collection name, the set name, and the matrix name. For example, the full name of the matrix `BCSSTK14` in the set `BCSSTRUC2` from the Harwell-Boeing Collection is `Harwell-Boeing/bcsstruc2/bcsstk14`. Note that both set name and matrix name are in lower case.

```
> matrixdepot("Harwell-Boeing/bcsstruc2/bcsstk14", :get)
  \% Total     \% Received \% Xferd  Average Speed   Time     Time      Time  Current
                                  Dload  Upload   Total   Spent    Left  Speed
100  292k  100  292k    0     0  22635      0  0:00:13  0:00:13 --:--:-- 61144
download:/home/weijian/.julia/v0.4/MatrixDepot/data/mm/Harwell-Boeing/bcsstruc2
/bcsstk14.mtx.gz

> matrixdepot("Harwell-Boeing/bcsstruc2/bcsstk14", :read)
1806x1806 Symmetric{Float64,SparseMatrixCSC{Float64,Int64}}:
      1.93161e6  0.0    -1.02166e5  ...      0.0            0.0
      0.0        1.0     0.0                 0.0            0.0
     -1.02166e5  0.0     1.93147e6           0.0            0.0
 -35568.9        0.0     1.65787e5           0.0            0.0
     -1.06959e5  0.0    -1.06959e5           0.0            0.0
     -1.65835e5  0.0  35568.9        ...      0.0            0.0
   -717.845      0.0     0.0                 0.0            0.0
      0.0        0.0  88998.5                0.0            0.0
      0.0        0.0    -1.82865e6           0.0            0.0
      0.0        0.0     1.24988e5           0.0            0.0
      ...                                    ...
      0.0        0.0     0.0                 0.0            0.0
      0.0        0.0     0.0            1.06103e7     -5.25151e5
      0.0        0.0     0.0           -5.25151e5   -53434.0
```

```
        0.0         0.0         0.0     ...     1.06959e5     -1.65835e5
        0.0         0.0         0.0             0.0            0.0
        0.0         0.0         0.0             1.06959e5     35568.9
        0.0         0.0         0.0            -816518.0       1.21311e7
        0.0         0.0         0.0             4.55624e7      8.15266e5
        0.0         0.0         0.0     ...     8.15266e5      5.27942e8
```

We recommend downloading matrices from the University of Florida Sparse Matrix Collection when there is a choice, because almost every matrix from Matrix Market is included in it.

## CONCLUDING REMARKS

Matrix Depot follows in the footsteps of earlier collections of matrices. Its novelty is threefold. First, it is extensible by the user, and so can be adapted to the user's needs. In doing so it facilitates experimentation, and in particular makes it easier to do reproducible research. Second, it combines several existing test matrix collections, namely Higham's Test Matrix Toolbox, Hansen's regularization problems, and the University of Florida Sparse Matrix Collection, in order to provide both parametrized test matrices and real-life sparse matrix data in a single framework. Third, it fully exploits the Julia language. It uses multiple dispatch to help provide a simple interface and, in particular, to allow matrices to be generated in any of the numeric data types supported by the language. Matrix Depot therefore anticipates the development of intrinsic support in Julia for computations with `BigFloat` and other data types.

Matrix Depot has been in development since 2014. It is an open source project (https://github.com/weijianzhang/MatrixDepot.jl) hosted on GitHub and is available under the MIT License. A first release was announced in December 2014. Matrix Depot `v0.5.5` is the latest official release and consists of around 3,000 lines of source code, with test coverage of 98.91% according to Codecov (https://codecov.io/). From GitHub traffic analytics, we learn that Matrix Depot has 40–70 unique downloads (unique cloners) every month. Matrix Depot also benefits the development of other Julia packages. LightGraphs (https://github.com/JuliaGraphs/LightGraphs.jl), an optimized graph package for Julia, for example, has embedded Matrix Depot as its database.

We built Matrix Depot to facilitate the development and testing of matrix (and other) algorithms in Julia. and we will continue to develop Matrix Depot by introducing new test matrices and integrating other test collections.

## ACKNOWLEDGEMENTS

The authors are grateful to Jiahao Chen (MIT), Stefan Güttel (The University of Manchester), and Tim Davis (Texas A & M University) for suggestions, and to Per Christian Hansen for allowing us to incorporate problems from Regularization Tools.

### Funding

The work of Higham was supported by European Research Council Advanced Grant MATFUN (267526) and Engineering and Physical Sciences Research Council grant EP/I01912X/1. The funders had no role in study design, data collection and analysis, decision to publish, or preparation of the manuscript.

### Grant Disclosures

The following grant information was disclosed by the authors:
European Research Council Advanced Grant MATFUN: 267526.
Engineering and Physical Sciences Research Council grant: EP/I01912X/1.

### Competing Interests

Nicholas J. Higham is an Academic Edtor for PeerJ Computer Science.

### Author Contributions

- Weijian Zhang conceived and designed the experiments, performed the experiments, analyzed the data, contributed reagents/materials/analysis tools, wrote the paper, prepared figures and/or tables, performed the computation work, reviewed drafts of the paper.
- Nicholas J. Higham conceived and designed the experiments, analyzed the data, contributed reagents/materials/analysis tools, wrote the paper, performed the computation work, reviewed drafts of the paper.

### Data Availability

Matrix Depot: https://github.com/weijianzhang/MatrixDepot.jl.

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
