# Peer review of "Matrix Depot: an extensible test matrix collection for Julia"

_PeerJ Computer Science, doi:10.7717/peerj-cs.58_

## Round 0.1 · original submission · Minor Revisions

The manuscript has now been seen by two Reviewers, who agree that it passes all criteria for publication in PeerJ CS. I would ask the Authors to consider the careful feedback by Reviewer 1, and submit a minor revision of the manuscript that addresses those observations and suggestions that the Authors judge can be carried out in the context of a minor revision.

·

Basic reporting

Overall, it's an excellent paper and package. Please consider
the following issues in the final published paper and code.

Detailed comments:

Abstract:
The three goals are excellent: (1) the user can add their own
test problems and use the same framework, (2) one framework
with (2a) parameterized matrices (which can be generated),
(2b) regularization problems, and (2c) matrices from a mix
of applications and (3) uses Julia and its support for multiple
numeric types.

I don't understand, at least at the abstract, why (2a) and
(2b) are different. It seems to me that the regularization
problems are simply a special case of the generatable, parameterized
matrices. That is, (2a) and (2b) seem the same to me.

(2b) just seems to be a particular set of generatable matrices,
which also have the right-hand side b and given solution x.
But other matrices in (2a) could presumably give A,b, and x.
Likewise, many matrices in the UF collection also have both A and b.

Why treat (2b) as anything special? It seems to me that your
matrix depot will break when someone wants to include matrix
problems from another domain that include more than just
r.A, r.b, and r.x (why not r.eigs, ...? r.coord for
matrices arising from a 2D/3D discretization, for which you want
to keep the 2D or 3D coordinates of each node/row/col of the matrix?).

For issue (3), I'm interested to know what you will do with a matrix
from (2c) the mix of applications. Suppose you have a matrix given to
you with a fixed precision. What would it mean to (say) create
a quad precision version of that? When the matrix only has
64 bit floating-point values at best?

Similarly, what if you ask for a matrix from the UF
collection in rational form? Do nearest rational representations
get created? That sounds like it could be numerically hazardous,
unless you use a lossless translation from double precision to
a ratio of integers. The rational format would be really ugly
for 0.3333333333 which might be one epsilon away from 1/3.

As an aside, I will be renaming the "University of Florida
Sparse Matrix Collection" to "The SuiteSparse Matrix Collection
(formerly known as the University of Florida Sparse Matrix
Collection)." The web site will move from its current ufl.edu
address to a new one at tamu.edu. The content of the collection
will remain the same. Can Julia be easily modified, perhaps
by the user, to reflect a change in URL to the SuiteSparse aka
UF collection? You might want to make a note of the upcoming
change, in citation [8] perhaps. I will likely be able to
continue to mirror the collection at both sites, however.

Perhaps a comment is useful: "In case the URL of the
external collection changes, you can give MatrixDepot the new
URL by ...".

Does the Julia download of matrices from the UF collection
preserve the meta data in each UF Problem struct?
Such as "notes"? "title"? "author"? etc? Or does that get
lost? Some Problems have problem-specific meta data.
For example, the IMDB movie database has names for each
row and column of the matrix. Another Problem (Moqri/MISKnowledgemap) is a set of documents, and the abstract of each document is included in the Problem. (matrix ID 2663).

Having read through the paper, I see that my questions
are unanswered.

It would be important to preserve the metadata for matrices
in the SuiteSparse aka UF Collection. Some information
("notes", "author", "title", "id", and others) are in
all matrices. Other data is problem-specific.

I see you have a way of defining r.x for a regularization
problem. Why not extend this to return all the metadata

r = maxtrixdepot ("HB/arc130") ;

which gives:

r.A the sparse matrix
r.title the title in the SuiteSparse aka UF collection
r.b the right hand side (if it exists)
r.notes the notes about the matrix
r.kind the kind of matrix

and so on. If a Problem has extra, non-standard information,
it goes in the 'aux' field, which could be either:

r.aux.c
r.aux.lo
r.aux.hi
r.aux.z0
r.aux.coord
r.aux.actorname
r.aux.nodename
r.aux.whateveryoufindgoeshere

or perhaps a flattened structure is fine too:

r.aux_c
r.aux_lo
r.aux_hi
r.aux_z0
r.aux_coord
r.aux_actorname
r.aux_nodename
r.aux_whateveryoufindgoeshere

The r.aux.c, lo, hi, and z0 are used for linear programming
problems. r.aux.coord is for 2D or 3D coordinates, if the
problem comes from a 2D/3D discretization. I don't have
many matrices with 2D/3D coordinates, but that info is very
important to some methods.

All of my linear programming problems have c, lo, hi,
and z0. All of my model reduction problems have yet another
set of common 'aux' fields. And so on, just like your
regularization problems, but extended to more than just one
class.

Since you can already have r.A, r.x, and r.b, it seems like
it would not be hard to extend this to r.anything.

Can that extension be done dynamically, without the need
to modify Julia?

I'd really like to see a format that can accommodate more
than just regularization problems.

In my collection, I keep each matrix and all its meta
data in 3 formats (MATLAB, Matrix Market, and Rutherford Boeing).
When I generate the 3 formats, I ensure that each format
contains exactly the same information, down to the very
last bit. There is no O(eps) variation between the matrix
values, for example. I preserve all the metadata too.
The Julia Matrix Depot would drop the meta data.

What do you do with the explicit zeros that are present in
some sparse matrices? Those are in the *.mtx file, for
the Matrix Market format. I assume you download that copy.
Are they preserved in Julia? This is a minor nuance, but
an important one. You don't have to address this in the
paper; just in the code. I preserve them for MATLAB by
including another binary sparse matrix, Problem.Zeros,
which is 1 in the (i,j) position if the given matrix has an
expliticly provided entry whose value is zero, in that position.
Does that information get preserved? It's important to keep it;
that structure is important to the matrix problem.

The overall gist of my review is this. The matrix depot for Julia is user-extendible, which is great. Can the *content* of each matrix problem also be extendible, to include extra meta data that is specific to each problem? Can this be done without needing to rewrite the matrix depot interface? If so, the 'regularization problem' becomes just one in a host of possible special problems. Is that possible in Julia? I think such a framework would greatly strengthen the power of this matrix depot.

Overall, it's an excellent paper and package, and I look forward to seeing the final paper. Please consider these issues in the final published paper and code.

Experimental design

no applicable.

Validity of the findings

no comments.

Additional comments

I wrote a short MATLAB script that queries each problem in the UF Collection. It lists all the various top-level fields, and the kinds of aux fields that are currently in use. I can't seem to attach it to this review, however. Here it is below:

clear
index = UFget ;
nmat = length (index.nrows)

fields = { } ;
counts = [ ] ;

for id = 1:nmat

% get all the field names from the Problem
Problem = UFget (id, index)
f = fieldnames (Problem) ;
if (isfield (Problem, 'aux'))
auxfields = fieldnames (Problem.aux)
for i = 1:length (auxfields)
auxfields {i} = [ 'aux.' auxfields{i} ] ;
end
else
auxfields = { } ;
end
f = [f ; auxfields]

% set union, but keep the counts
for i = 1:length (f)
k = find (ismember (fields, f {i})) ;
if (isempty (k))
fields = [fields ; f{i} ] ;
counts = [counts 0] ;
k = length (fields) ;
end
counts (k) = counts (k) + 1 ;
end

end

save getallfields fields counts

fprintf ('\nTotal number of matrices: %d as of %s\n', nmat, date) ;
for i = 1:length (fields)
fprintf ('%5d : %s\n', counts (i), fields {i}) ;
end

Here is the output, where I sorted the results as well:

Total number of matrices: 2757 as of 22-Jan-2016

Every problem has these fields:

2757 : title
2757 : name
2757 : kind
2757 : id
2757 : ed
2757 : date
2757 : author
2757 : A

Then each problem may have these fields. For instance,
1717 of them have Problem.notes. These fields are sometimes
vectors or matrices (sparse or dense), scalars, or text.

1717 : notes
889 : b
789 : aux
417 : Zeros
346 : aux.c
342 : aux.z0
342 : aux.lo
342 : aux.hi
187 : aux.coord
101 : aux.nodename
95 : aux.rowname
91 : aux.mapping
63 : aux.B
62 : aux.C
60 : aux.E
50 : x
50 : aux.population
50 : aux.area
31 : aux.A
27 : aux.b
16 : aux.cname
14 : aux.M
12 : aux.G
11 : aux.iv
10 : aux.Gname
6 : aux.solution
6 : aux.smooth_number
6 : aux.pubyear
6 : aux.gcs
6 : aux.factor_base
5 : aux.colname
4 : aux.year
4 : aux.nodevalue
4 : aux.K
4 : aux.cluster
3 : aux.TSSOAR
3 : aux.partition
3 : aux.Mzeros
3 : aux.guess
3 : aux.Gcoord
3 : aux.date
2 : aux.Zeros
2 : aux.w0
2 : aux.subcat
2 : aux.shift
2 : aux.S
2 : aux.country
2 : aux.class
2 : aux.category
2 : aux.cat
2 : aux.b2
2 : aux.b1
2 : aux.appyear
1 : aux.Year
1 : aux.Varnoldi
1 : aux.Vansys
1 : aux.Topics
1 : aux.T
1 : aux.t
1 : aux.Source
1 : aux.servicecode
1 : aux.service
1 : aux.roots
1 : aux.rootname
1 : aux.Q
1 : aux.ps
1 : aux.PIN_class
1 : aux.phdyear
1 : aux.nodesource
1 : aux.nodes
1 : aux.nodecode
1 : aux.MovieBacon
1 : aux.Label
1 : aux.Keywords
1 : aux.key
1 : aux.KevinBacon
1 : aux.inbook
1 : aux.id
1 : aux.elements
1 : aux.edgecode
1 : aux.Day
1 : aux.D
1 : aux.concreteness
1 : aux.code
1 : aux.closedform
1 : aux.CiteCnt
1 : aux.Authors
1 : aux.Atop
1 : aux.Aside
1 : aux.ActorBacon
1 : aux.Abstract
1 : aux.Abottom

I'd like to take a 2nd pass at the revised paper, so I can see how you propose to take into account the presence of these extra components to the matrix problems.

Reviewer 2 ·

Basic reporting

The manuscript passes all essential criteria, excellent exposition, sufficient introduction and background. The subject itself is useful to a broader range of researchers for the years to come.

Experimental design

Pass

Validity of the findings

Pass

---

## Round 0.2 · accepted · Accept

The submitted revision improves over the previous version of the manuscript by incorporating most of the suggestions of Reviewer 1 and addressing several minor issues. The revised manuscript is thus acceptable for publication in its present form.